# A Multispectral UAV Imagery Dataset of Wheat, Soybean and Barley Crops in East Kazakhstan

Almasbek Maulit [1,*] , Aliya Nugumanova [2,*] , Kurmash Apayev [3] , Yerzhan Baiburin [1] and Maxim Sutula [4]

1 Laboratory of Digital Technologies and Modeling, Sarsen Amanzholov East Kazakhstan University, Ust-Kamenogorsk 070004, Kazakhstan
2 Big Data and Blockchain Technologies Research Innovation Center, Astana IT University, Astana 010000, Kazakhstan
3 Department of Information Technologies, D. Serikbayev East Kazakhstan Technical University, Ust-Kamenogorsk 070000, Kazakhstan
4 Laboratory of Biotechnology and Plant Breeding, National Center for Biotechnology, Astana 010000, Kazakhstan
* Correspondence: maulit.almas@gmail.com (A.M.); a.nugumanova@astanait.edu.kz (A.N.)

**Abstract:** This study introduces a dataset of crop imagery captured during the 2022 growing season in the Eastern Kazakhstan region. The images were acquired using a multispectral camera mounted on an unmanned aerial vehicle (DJI Phantom 4). The agricultural land, encompassing 27 hectares and cultivated with wheat, barley, and soybean, was subjected to five aerial multispectral photography sessions throughout the growing season. This facilitated thorough monitoring of the most important phenological stages of crop development in the experimental design, which consisted of 27 plots, each covering one hectare. The collected imagery underwent enhancement and expansion, integrating a sixth band that embodies the normalized difference vegetation index (NDVI) values in conjunction with the original five multispectral bands (Blue, Green, Red, Red Edge, and Near Infrared Red). This amplification enables a more effective evaluation of vegetation health and growth, rendering the enriched dataset a valuable resource for the progression and validation of crop monitoring and yield prediction models, as well as for the exploration of precision agriculture methodologies.

**Keywords:** multispectral UAV imagery; NDVI; remote sensing; crop monitoring; yield prediction; precision agriculture

## 1. Summary

In recent years, unmanned aerial vehicles (UAVs) have emerged as a prevalent tool for acquiring agricultural crop data [1–5]. These vehicles are outfitted with cameras that obtain high-resolution multispectral images, which are instrumental in tracking crop health, growth, and development. The benefits of UAV-based data collection compared to satellite-based collection include higher resolution, daily data availability upon request, and the ability to capture images in cloudy weather conditions. In this work, the Phantom 4 multispectral drone is used, which captures data at a resolution of 3 cm and on-demand imaging, compared to the well-known satellite Sentinel-2, which provides lower resolutions of 10, 20, or 60 m in the visible spectrum and data only every 5 days. UAVs have transformed agriculture by allowing affordable and efficient data collection, enhancing crop management, and facilitating informed decision-making across vast areas. As a result, numerous research studies are focusing on utilizing UAVs for crop monitoring.

In [6], Sentinel-2 satellite data and multispectral UAV data for crop monitoring in Northeastern Germany are compared. The authors demonstrate that UAV data has a stronger correlation with agronomic variables, specifically with a 6.3% increase for wheat and a 22.2% improvement for barley. They recommend using UAV data for decision-making, given its superior performance in calculating vegetation indices. In [7], utilizing an UAV with a commercial digital camera to estimate SPAD values indicating naked barley leaf health is explored. The authors offer useful insights for budget-conscious farmers by improving the correlation between image-derived vegetation indices and SPAD values. In [8], the focus is on the challenge of soil in UAV images of crops. The authors identify the most sensitive vegetation indices to refine wheat trait predictions by removing soil pixels from the distribution of index values. In [9], the potential of a multispectral camera on an UAV is examined for guiding in-season nitrogen management in wheat production. The authors propose a modified sufficiency index algorithm to provide variable rate nitrogen recommendations, resulting in an improved harvest index and reduced nitrogen input without affecting yield.

The works discussed above investigate different aspects of using UAVs for crop monitoring, with each work utilizing unique data analysis methods. However, data collection methods, such as organizing flight missions, conducting flights, and image processing, are generally standard. Nevertheless, the development of innovative precision agriculture methods would not be possible without undertaking these time- and labor-intensive data collection steps. In this regard, open datasets containing raw UAV data from the field are crucial for researchers who may lack the capability to obtain such data.

A focused search was executed using the Advanced Search Query Builder on the Web of Science platform to identify pertinent dataset publications in four authoritative data journals, namely "Data" (MDPI), "Data in Brief" (Elsevier), "Earth System Science Data" (Copernicus), and "Scientific Data" (Nature Research). The search was centered around the topics "UAV" and "crop" and generated the following outcomes:

- The journal "Data" contains just one relevant data descriptor article [10], which showcases a dataset of UAV RGB images captured over a pistachio orchard.
- The journal "Data in Brief" features seven relevant articles [11–17], including one that pertains to UAV images of a cotton field [11], another that focuses on UAV data for avocado classification [12], two that present UAV images obtained over a vineyard [13,14], and one that presents plant and soil data for forage crops [15]. Additionally, there is an article that showcases UAV RGB images of soybean crops [16] and another that features hyperspectral imagery of potato cultivation [17].
- The journals "Earth System Science Data" and "Scientific Data" each contain only one article providing data on grassland aboveground biomass on the Qinghai-Tibet Plateau [18] and forest ecosystem 3D perception in the Hainich-Dün region in Germany [19], respectively.

Based on the results, it appears that there is currently a shortage of openly available datasets specifically focused on using UAVs for crop monitoring. While there is one relevant study [16] that focused on soybean culture, the images were collected in RGB format and were intended for pest recognition purposes. To address this gap, the current study presents an imagery dataset of wheat, soybean, and barley crops that were captured in Eastern Kazakhstan in 2022 using a DJI Phantom 4 UAV equipped with a multispectral camera. The imagery embodies the progression of crop growth at various phenological stages, encompassing emergence, tillering, heading, and ripening. The dataset features two primary components: (1) the raw imagery, directly collected by the UAV and including images in both TIF and JPEG formats, and (2) the processed imagery, available solely as TIF files, which have been enhanced by adding an NDVI band to the original TIF bands. It is expected that the dataset will be used to develop and test new algorithms and models for crop monitoring, disease detection, yield estimation, and other applications.

## 2. Data Description

The suggested UAV imagery dataset is comprised of eight compressed (zip) files grouped into two components. The first component of the dataset, raw imagery captured by the UAV, is represented by five zip files by the number of flight sessions, while the second component, processed orthomosaic imagery categorized by crops, is represented by the remaining three zip files by the number of crop types grown. Table 1 offers an overview of the mentioned files. In the dataset, crops are not evenly represented, with 12 wheat plots, 12 barley plots, and just 3 soybean plots in the study. However, as mentioned earlier, all plots have the same area of 1 hectare, regardless of the crop grown. Detailed descriptions of Component one and Component two are provided in Sections 2.1 and 2.2, respectively.

**Table 1.** Summary of the compressed files in the UAV imagery dataset.

| No. | File Name | Size | Number of Files | Description |
|-----|-----------|------|-----------------|-------------|
| 1 | flight_session_01.zip | 17.7 Gb | 8277 | |
| 2 | flight_session_02.zip | 17.4 Gb | 7957 | Component 1: raw aerial |
| 3 | flight_session_03.zip | 16.7 Gb | 7721 | images from the corresponding |
| 4 | flight_session_04.zip | 17.5 Gb | 7267 | UAV flight session |
| 5 | flight_session_05.zip | 15.8 Gb | 7155 | |
| 6 | barley.zip | 12.7 Gb | 59 | Component 2: processed |
| 7 | wheat.zip | 11.8 Gb | 60 | orthomosaic images of plots, categorized by crop type |
| 8 | soybean.zip | 2.34 Gb | 12 | (barley, wheat, soybean) |

### 2.1. The Raw Imagery Captured by the UAV

In Component one, every flight-session zip file features a two-tiered folder organization. Initially, raw images are arranged according to their capture dates. Within each date folder, images are further sorted into subfolders named FPLAN folders according to the flight plans. The number of FPLAN folders ranges between 6 and 14, contingent on the specific flight mission configuration for the respective day. For instance, Figure 1 illustrates the contents of the 102FPLAN folder, corresponding to the date 2022-06-08 and originating from the second flight session. As shown in Figure 1, each FPLAN subfolder contains four system files and numerous individual images captured by the UAV in photography mode. Each image is available in two formats: RGB format (represented by a JPEG file) and monochrome format (represented by five TIF files, which depict the image in various bands). The final digit in the name of the TIF file indicates the corresponding band: 1—blue, 2—green, 3—red, 4—red-edge, and 5—near-infrared red.

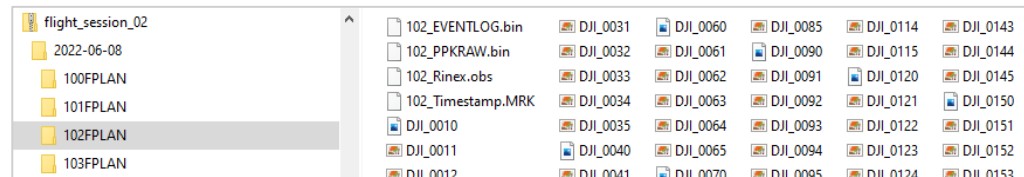

**Figure 1.** Depiction of the folder structure within the **flight_session_02.zip** file showcasing the contents of the **2022-06-08/102FPLAN** subfolder.

Each image file exhibits a pixel size of 1600 × 1300 and a resolution of 3 cm per pixel. The drone's multispectral camera automatically tagged these images using the EXIF format. Specifically, geotags encompass the latitude and longitude coordinates of the image's lower-left point, as well as the altitude above sea level. Figure 2 showcases two images of the same plot, taken by the UAV on 21 June during the 75% heading phase (Figure 2a) and on 11 July during the phase of milky rape (Figure 2b).

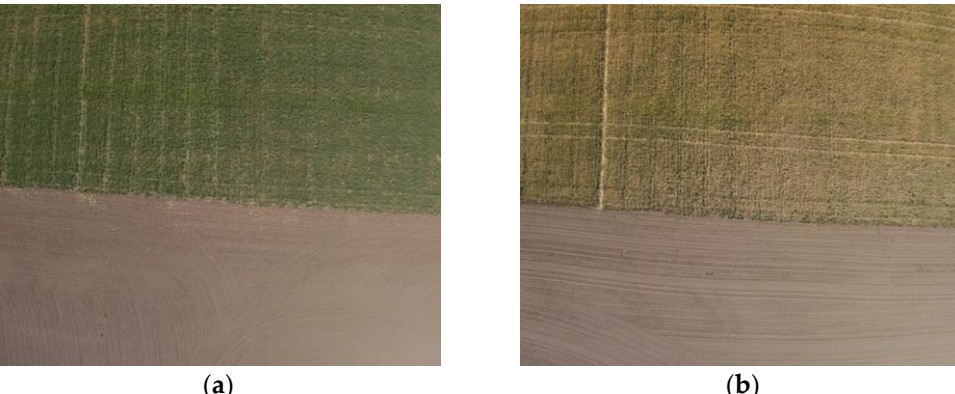

(**a**)                                                                        (**b**)

**Figure 2.** A comparison of two images of the same plot. (**a**) on 21 June; (**b**) on 11 July.

### 2.2. The Processed and Enhanced UAV Imagery Dataset

In Component two, every zip file features a two-tiered folder organization. First, images are sorted based on their corresponding crop types. Inside each crop-type folder, images are then organized into plot-specific subfolders (see Figure 3). Consequently, each plot folder houses processed TIF images of the crop captured on different aerial photography dates. As previously noted, each TIF file comprises: (1) an orthomosaic formed by approximately 47–50 smaller, adjacent images captured by the UAV, collectively representing a 1 hectare experimental plot; (2) a collection of six bands, including five spectral bands and an additional band containing calculated NDVI values (see Figure 4).

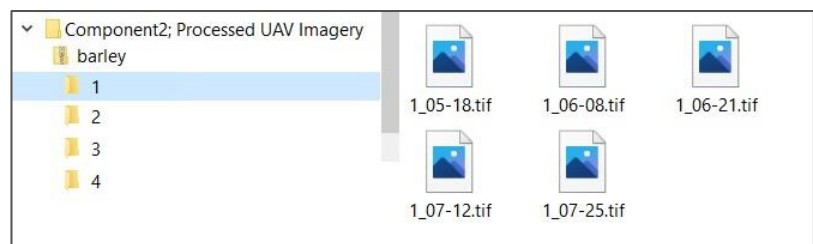

**Figure 3.** Depiction of the folder structure within the barley.zip file showcasing the contents of the plot 1 subfolder.

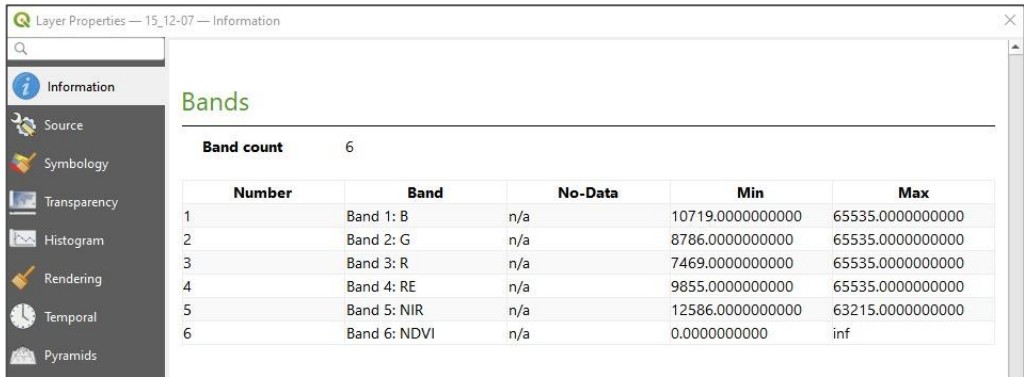

**Figure 4.** Orthomosaic of a 1-hectare experimental plot generated in Agisoft Metashape by merging adjacent images obtained by the UAV.

## 3. Methods

### 3.1. Experimental Site and Plots

The experimental site is situated near the city of Ust-Kamenogorsk in the East Kazakhstan region of the Republic of Kazakhstan, at coordinates 50°02′ N 82°35′ E, encompassing an area of 27 hectares. In 2022, three crops were cultivated: wheat, barley, and soy (see

Figures 5–7). Wheat varieties (Altai, Ulbinka 25, Nargiz, GVK 2120/3) were each grown on 1-hectare plots, totaling 12 plots with varying sowing rates. Barley varieties (Ilek 16; L 29; 339 A; B 2015) were similarly grown on 1-hectare plots, amounting to 12 plots with different sowing rates. Soybean (Birlik) was cultivated on 3 hectares, also with varying sowing rates. A brief description of the experimental plots and the cultivated crops is presented in Table 2.

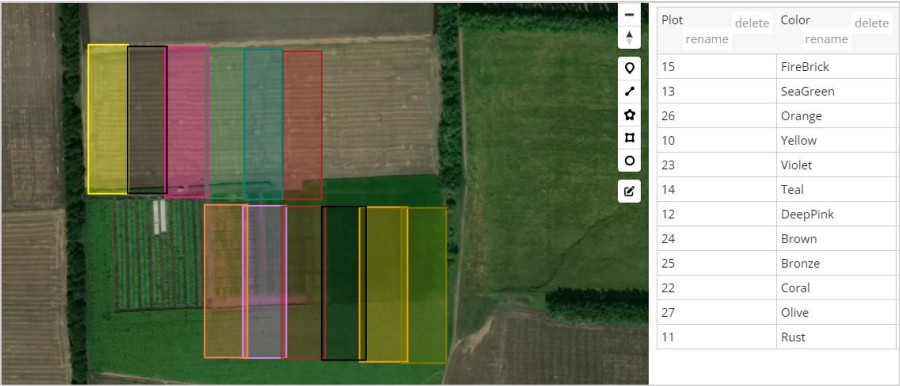

**Figure 5.** Location of wheat plots within the experimental site.

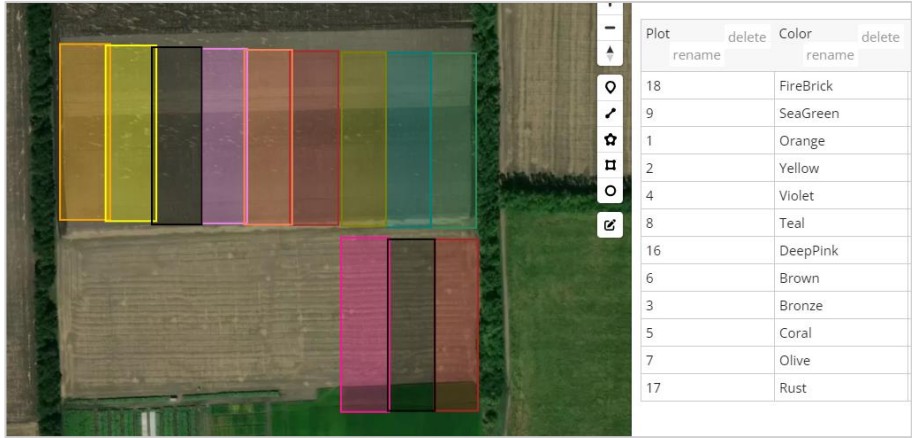

**Figure 6.** Location of barley plots within the experimental site.

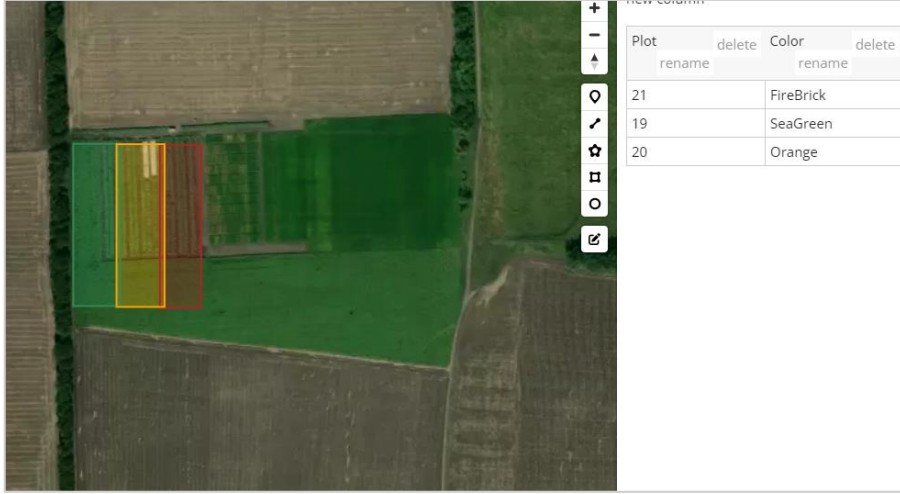

**Figure 7.** Location of soybean plots within the experimental site.

**Table 2.** Characteristics of cultivated crops on experimental plots.

| Plot ID | Crop Type | Crop Variety | Sowing Rate (kg/ha) | Date of Sowing |
|---|---|---|---|---|
| 1 | barley | Ilek 16 | 280 | 4 May 2022 |
| 2 | barley | Ilek 16 | 260 | 5 May 2022 |
| 3 | barley | Ilek 16 | 240 | 5 May 2022 |
| 4 | barley | L 29 | 280 | 5 May 2022 |
| 5 | barley | L 29 | 260 | 5 May 2022 |
| 6 | barley | L 29 | 240 | 6 May 2022 |
| 7 | barley | 339 A | 280 | 6 May 2022 |
| 8 | barley | 339 A | 260 | 6 May 2022 |
| 9 | barley | 339 A | 240 | 6 May 2022 |
| 10 | wheat | Altay | 240 | 7 May 2022 |
| 11 | wheat | Altay | 220 | 8 May 2022 |
| 12 | wheat | Altay | 200 | 8 May 2022 |
| 13 | wheat | Ulbinka | 240 | 13 May 2022 |
| 14 | wheat | Ulbinka | 220 | 13 May 2022 |
| 15 | wheat | Ulbinka | 200 | 13 May 2022 |
| 16 | barley | B 2015 | 280 | 7 May 2022 |
| 17 | barley | B 2015 | 260 | 7 May 2022 |
| 18 | barley | B 2015 | 240 | 7 May 2022 |
| 19 | soybean | Birlik | 120 | 21 May 2022 |
| 20 | soybean | Birlik | 100 | 21 May 2022 |
| 21 | soybean | Birlik | 80 | 21 May 2022 |
| 22 | wheat | Nargiz | 240 | 13 May 2022 |
| 23 | wheat | Nargiz | 220 | 13 May 2022 |
| 24 | wheat | Nargiz | 200 | 13 May 2022 |
| 25 | wheat | GBK 2120/3 | 240 | 14 May 2022 |
| 26 | wheat | GBK 2120/3 | 220 | 14 May 2022 |
| 27 | wheat | GBK 2120/3 | 200 | 14 May 2022 |

Situated in the Rudno-Altai structural-facies region and the southern part of the temperate climate zone, the experimental site is close to the pole of continentality. With a moderately cool climate and an average annual precipitation of 597 mm, rainfall occurs even during the driest periods. The site features typical chernozem soil, with 10.1–15 mg/kg nitrogen, 31–45 mg/kg mobile phosphorus, and 301–400 mg/kg exchangeable potassium content. This fertile soil type and temperate climate foster diverse vegetation in the region, such as grasslands, forests, and cultivated farmlands.

### 3.2. UAV Platform and Mission

The DJI Phantom 4 multispectral UAV is equipped with a 6 × 1.29-inch CMOS multispectral camera that captures visible RGB and multispectral images using five monochromatic sensors (Figure 8). The camera has a 62.7° field of view, 5.74 mm focal length, and f/2.2 aperture. It produces images with a resolution of 1600 × 1300 pixels in JPEG and TIF formats. The UAV can reach a maximum horizontal velocity of 50–58 km/h, ascent speeds of 5–6 m/s, and a descent speed of 3 m/s. It has a maximum wind resistance of 10 m/s and an operational altitude of up to 6000 m. The drone features obstacle detection and an infrared system for safe navigation.

The DJI TimeSync Accurate Data Acquisition System ensures real-time geolocation data for each image, offering centimeter-level precision. Powered by LiPo PH4 15.2V intelligent flight batteries, the UAV weighs 1487 g and provides a maximum flight time of approximately 27 min. It uses GPS + BeiDou + Galileo or GPS + GLONASS + Galileo positioning systems for accurate navigation. A total of five flight sessions were carried out by the UAV to capture the phenological development of the plants at specific milestones. To address practical limitations in terms of human resources and/or weather conditions, all five flight sessions were spread over two consecutive days. Specifically, during the first day of these two-day sessions, the UAV covered approximately half of the available plots, with the remaining plots being captured on the second day of the session. All the flight

sessions (excluding the first one) took place between 10:00 a.m. and 2:00 p.m. local time to minimize filming distortions. Tables 3 and 4 provide information about the schedule of the flight sessions and conditions under which the images were captured, respectively.

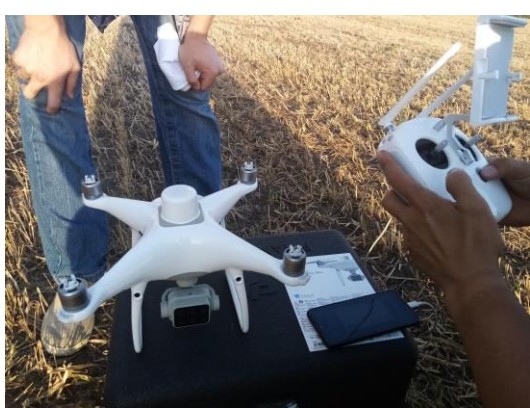

**Figure 8.** The DJI Phantom 4 multispectral UAV at the experimental site.

**Table 3.** Schedule of the conducted flight sessions, including the date, duration, and corresponding phenological stage (for wheat and barley).

| Flight Session | Milestone (Corresponding Phenological Stage) | Date | Covered Plots |
|---|---|---|---|
| 01 | 75% emergence | 17 May 2022<br>18 May 2022 | 15<br>13 |
| 02 | Tillering | 8 June 2022<br>9 June 2022 | 14<br>7 |
| 03 | 75% heading | 21 June 2022<br>22 June 2022 | 8<br>7 |
| 04 | Milky ripe | 11 July 2022<br>12 July 2022 | 9<br>6 |
| 05 | Waxy ripe | 25 July 2022<br>26 July 2022 | 8<br>7 |

**Table 4.** The conditions under which the images were captured.

| Date of 2022 | Local Time | Sun Elevation Angle, ° | Sun Azimuth Angle, ° | Temperature, °C | Wind Speed, m/s | Cloud Coverage |
|---|---|---|---|---|---|---|
| 17 May | 13:00–17:00 | 54.25–19.09 | 219.66–277.9 | 27.7–13.7 | 1–2 | clear |
| 18 May | 11:00–14:30 | 59.16–42.85 | 167.61–246.97 | 30.2–22.0 | 1–2 | clear |
| 8 June | 09:30–12:30 | 54.22–60.39 | 128.31–209.1 | 26.8–27.2 | 2–3 | clear |
| 9 June | 09:30–12:30 | 54.26–60.48 | 128.17–209.07 | 30.6–31.0 | 1–2 | clear |
| 21 June | 09:30–12:30 | 54.33–61.11 | 126.89–208.26 | 32.7–24.0 | 5–6 | clear |
| 22 June | 09:30–12:30 | 54.3–62.65 | 126.82–163.62 | 34.9–35.2 | 3–4 | clear |
| 11 July | 10:30–12:30 | 59.15–60.05 | 149.13–205.7 | 30.7–30.9 | 2–3 | clear |
| 12 July | 09:30–11:30 | 52.57–61.84 | 127.08–177.01 | 31.3–29.7 | 4 | clear |
| 25 July | 10:00–12:30 | 53.95–57.72 | 138.91–203.88 | 24.6 –24.8 | 2–3 | clear |
| 26 July | 09:30–11:00 | 50.3–58.42 | 129.03–163.19 | 27.5–29.1 | 2–3 | clear |

Shooting plans were created using the DJI GS Pro app (version: 2.0.17), with a horizontal speed of 5 m/s and a flight altitude of 57 m above ground level. This setup allowed for a theoretical average ground resolution of 3.0 cm/px. The camera angle was set at 90°, side overlap ratios at 75%, and front overlap ratios at 60%. The camera was configured

to "Single Shot" in "Still Image Mode", with the highest possible image quality selected (1600 pixels × 1300 pixels). All photographs were automatically geotagged by the aircraft's multispectral camera in EXIF format, using the latitude/longitude (WGS84) coordinate system. To optimize battery usage, experimental plots were combined into larger sections. Figure 9 shows a flight mission as visualized using: (Figure 9a) the DJI GS Pro app; (Figure 9b) the precision agriculture framework specifically designed for this research project.

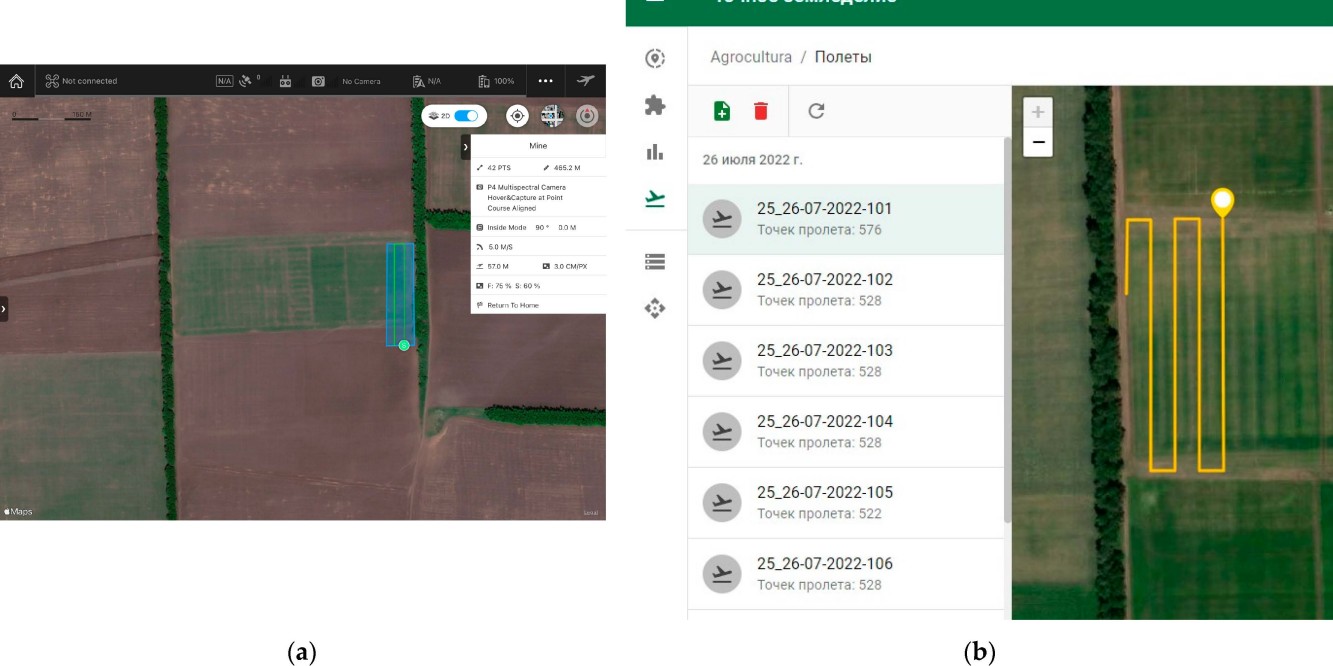

(**a**)　　　　　　　　　　　　　　　　　　　　　(**b**)

**Figure 9.** UAV flight mission: (**a**) A visualization with DJI GS Pro; (**b**) A visualization using the precision agriculture framework.

### 3.3. Processing the UAV Imagery Dataset

For all images, a photogrammetric reconstruction process was executed, resulting in the generation of a point cloud, digital elevation model (DEM), and an orthomosaic. The Agisoft Metashape Professional v1.8.3 software was employed to process the captured images, adhering to the software provider's recommendations [20]. Initially, the photographs were aligned with high precision, limited to 40,000 key points and 4000 connection points, followed by a camera optimization process. Subsequently, a medium-quality dense cloud was created with depth filtering disabled. Default settings were applied for all remaining parameters.

Thus, a high-quality digital elevation model (DEM) and an orthomosaic were generated (Figure 10). A DEM image provides a three-dimensional representation of the terrain and visualizes the elevations and contours of the land. An orthomosaic image enables users to orient themselves regarding the relative position of the plots in space. As depicted in Figure 10b, the field is partitioned into nine large plots, each of which is further divided into three smaller plots, resulting in a total of 27 plots. The distance between the larger sections is 2 m, and the distance between the smaller sections is 50 cm. The export of orthomosaic considers all spectral bands, resulting in a single multispectral orthomosaic that retains the same ranges as the original images. The methodology for processing multispectral data is like that of standard photographs, except for an additional step to select the primary channel after incorporating all images into the project.

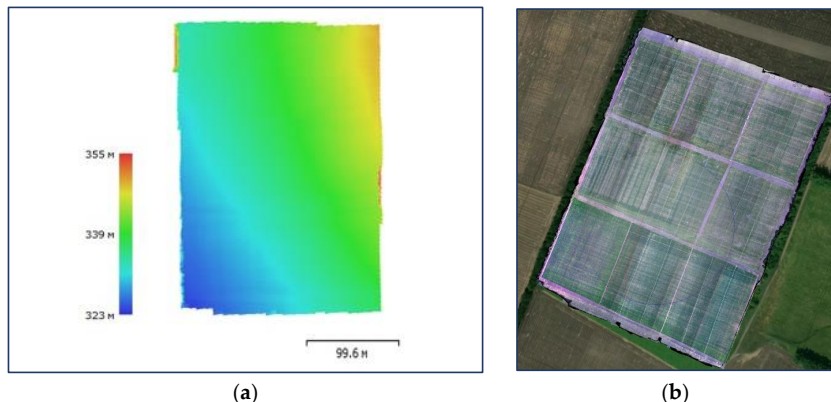

(**a**)                                                          (**b**)

**Figure 10.** (**a**) Digital elevation model representation; (**b**) Orthomosaic generated during the photogrammetric process.

### 3.4. Enhancing the UAV Imagery Dataset

Orthomosaic TIF files were imported into a Python application and processed using the Rasterio library. During this stage, the primary objects for processing were the spatial layers or rasters, which are georeferenced images. In other words, each raster contains a numeric pixel array for the image in its respective band, as well as metadata defining a rectangular extent corresponding to a specific spatial coordinate system. By utilizing the matrices from bands 3 and 5, a new NDVI value matrix was computed using the formula NDVI = (NIR − R)/(NIR + R), where NIR represents the near-infrared band, and R represents the red band. Once calculated, this matrix was added as the sixth layer in the original TIF file. To prevent confusion and enhance the usability and interpretability of the multispectral data, descriptive names were assigned to the layers in the application since the original source files only labeled the channels with numerical values. This step ensures that users can effortlessly identify and work with the various layers.

In Figures 11 and 12, two graphical representations are depicted, incorporating the application of Normalized Difference Vegetation Index (NDVI) layers. The initial figure's visualization reveals a chromatic scale where pixels with NDVI values close to 0.1 manifest as red, shifting to orange for values around 0.2 and progressively transitioning from red to green with increasing values. Notably, approximately 26.1% of the observed values are situated within the 0.4 range, suggesting a favorable outcome for the given survey period. In contrast, the subsequent figure exhibits a predominantly green appearance, with nearly 65% of the total plot area demonstrating NDVI values proximate to 0.4 and 35% demonstrating values around 0.5. The enhanced greenness of the second plot serves as an indicator of superior crop health in comparison to the first plot.

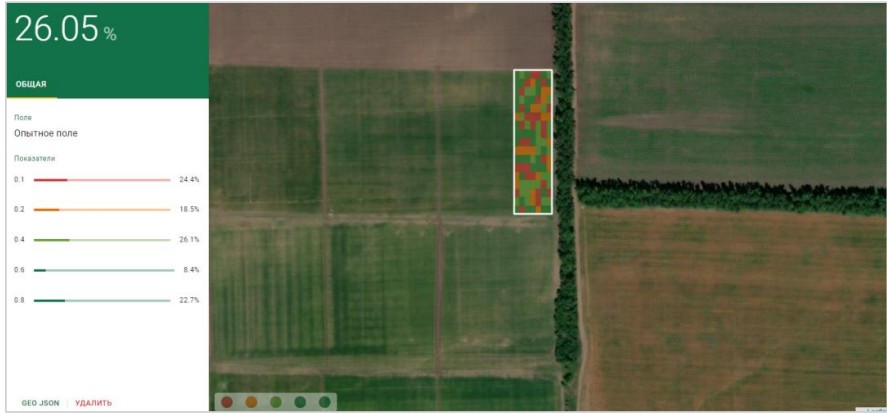

**Figure 11.** NDVI layer visualization showing a range of vegetation health values from weak to good, for the plot at the experimental site.

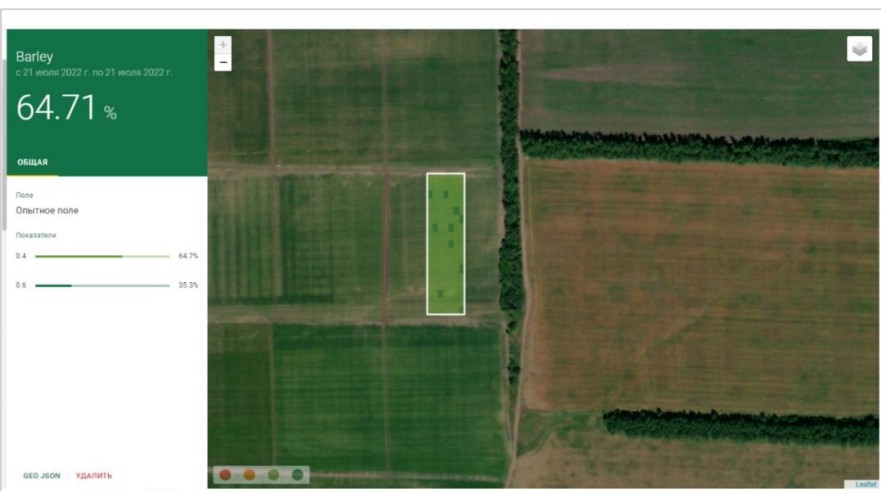

**Figure 12.** NDVI layer visualization with healthy vegetation dominance for the plot at the experimental site.

## 4. User Notes

The provided dataset meets the FAIR principles that stand for Findability, Accessibility, Interoperability, and Reusability [21]:

- The dataset is findable, as it has unique and persistent identifiers (DOIs).
- The dataset is accessible since it is hosted on Zenodo, an open-access repository that allows users to access and download the data.
- The dataset is interoperable since it is provided in commonly used formats (TIF, GeoTIF and JPEG) and includes an additional NDVI band for enhanced compatibility with other tools and platforms.
- The dataset is reusable since it includes metadata and is licensed under a Creative Commons Attribution 4.0 International License, which allows users to freely use and redistribute the data if they give appropriate credit.

In this section, two use cases for the created UAV imagery dataset are presented. The first use case outlines the process of reading and interpreting Component one of the dataset using Python in Google Colab. The second use case demonstrates the integration of Component two of the dataset into a framework for precision agriculture.

### 4.1. A Use Case for Utilizing Component One of the UAV Imagery Dataset

Component one of the dataset, i.e., the raw imagery, comprises TIF files, which lack georeferencing information, in contrast to GeoTIF files. While TIF files contain Exif metadata that offers details regarding the image capture location, this information only specifies the latitude and longitude of the bottom-left pixel within each image. To utilize this imagery for georeferencing and overlaying purposes with other spatial data, the location of every pixel must be determined. Specialized GIS software, such as Agisoft or QGIS, can automatically address this issue by employing the geospatial information from EXIF. However, if users wish to independently transform TIF into GeoTIF using Python, specific manipulations are necessary. This use case illustrates the conversation of a raw TIF image to GeoTIF format, utilizing the affine transformation. The complete code can be accessed through the provided link (https://colab.research.google.com/drive/1PTBHExWMbgxRkUreUFBaPrZmhOvAxYw3#scrollTo=2Iz7gsXQPJbc (accessed on 28 April 2023)). The following is an outline of the high-level algorithm:

1. Read the input image file and retrieve its width (*ImageWidth*) and height (*ImageHeight*) in pixels.
2. Extract GPS coordinates (latitude and longitude) of the bottom-left pixel from the image.

3. Convert the extracted longitude and latitude from Degrees/Minutes/Seconds format to decimal degree format as $X_{left\text{-}bottom}$ and $Y_{left\text{-}bottom}$, respectively using the following formula:

$$Decimal\ Degrees = (Degrees + Minutes\ /\ 60 + Seconds/3600) * Sign \tag{1}$$

where *Sigh* is +1 for North latitude or East longitude and −1 for South latitude or West longitude (in this case, sign is +1).

4. Compute the pixel size (in degrees) for both *X* and *Y* directions. Pixel size for *X* direction is calculated using the following formula:

$$Pixel\_size\_x = \frac{FlightResolution/100,000}{2\pi * EarthRadius * \cos(Y_{left\text{-}bottom})/360} \tag{2}$$

The Earth's radius (*EarthRadius*) is approximately 6371 kilometers (km). The flight resolution (*FlightResolution*) is defined by the application of the UAV and, in this case, is equal to 3 centimeters (cm) per pixel; to convert the resolution from cm to km, it is divided by 100,000. The latitude of the bottom-left pixel (*Latitude*) should be in decimal degrees. Therefore, the formula calculates the length of the Earth's circumference along one degree of longitude at a given latitude, considering that the Earth is not a perfect sphere and that its circumference varies with latitude. Pixel size for the *Y* direction is calculated using the following formula:

$$Pixel\_size\_y = \frac{FlightResolution/100,000}{111.3} \tag{3}$$

The value 111.3 represents the approximate number of kilometers along one degree of latitude. This value is relatively constant because lines of latitude are parallel and evenly spaced.

5. Create the geotransform matrix (Affine transformation) to convert the pixel coordinates of the image into geographic coordinates by:

   a. Translating the origin of the image coordinate system to the bottom-left corner of the image using the decimal degree longitude and latitude values of the bottom-left corner of the image. This ensures that subsequent transformations are performed relative to this corner, which is the reference point for geographic coordinates.

   b. Scaling the *X* (longitude) and *Y* (latitude) dimensions of the image by their respective pixel sizes in degrees. Scaling ensures that the transformation from pixel coordinates to geographic coordinates will be performed with the correct spacing between them.

6. Save the geotransform matrix as metadata within the GeoTIF file.

### 4.2. A Use Case for Integration of Component Two of the UAV Imagery Dataset in a Framework to Support Beginning Farmers

This research was conducted within a three-year project focused on developing a precision agriculture framework specifically designed for small-scale farming. The resulting technology and associated dataset were integrated into the framework as detailed below. To monitor a particular plot, an NDVI matrix was generated using satellite images (see Figure 13) or the UAV images. In areas with low NDVI values indicating plant growth stress, additional surveys were conducted by the UAV at elevations between 10 and 50 m to acquire a more accurate representation. The framework further provides users with supplementary services, such as humidity assessment, meteorological data analysis, and more. Currently, the platform is in the prototype development stage.

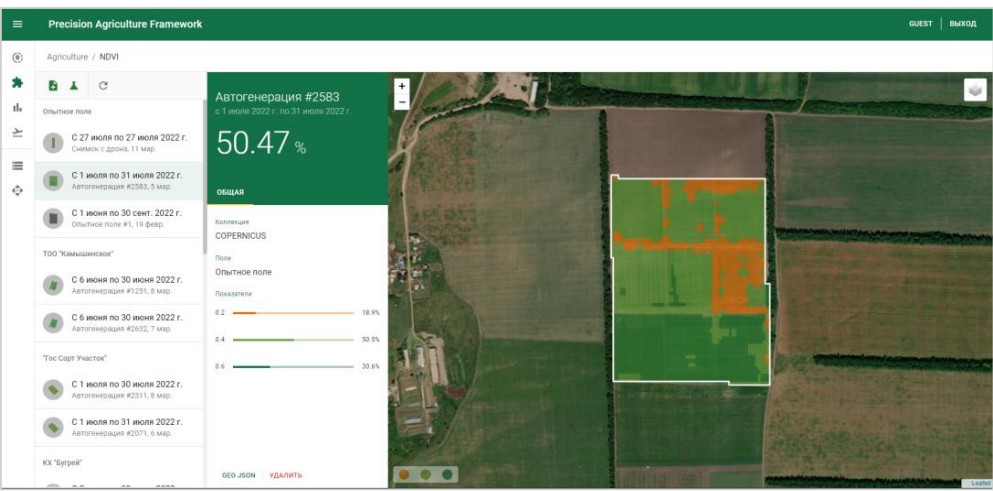

**Figure 13.** NDVI layer visualization for the plot.

Outlined below is the high-level algorithm for extracting NDVI values from a GeoTIF file and generating GeoJSON. After generating the GeoJSON, it can be integrated into a framework for further analysis or visualization.

1. Read the source GeoTIF file.
2. Read the sixth layer of the file, which represents the calculated NDVI values.
3. Extract the GPS coordinates and NDVI values for each pixel.
4. Create a polygon based on the GPS coordinates.
5. Create a GeoDataFrame from the polygon and NDVI values.
6. Generate GeoJSON from the GeoDataFrame.

**Author Contributions:** Conceptualization, A.N. and Y.B.; methodology, A.M.; software, K.A.; validation, A.M., A.N. and Y.B.; resources, M.S.; data curation, A.N. and A.M.; writing—original draft preparation, A.M. and A.N.; writing—review and editing, A.M. and A.N.; visualization, A.M. and K.A.; supervision, A.N. and M.S.; project administration, M.S.; funding acquisition, A.N. and A.M. All authors have read and agreed to the published version of the manuscript.

**Funding:** This research was funded by the Science Committee of the Ministry of Science and Higher Education of the Republic of Kazakhstan, grant number AP09259379.

**Institutional Review Board Statement:** Not applicable.

**Informed Consent Statement:** Not applicable.

**Data Availability Statement:** Link to the dataset: https://doi.org/10.5281/zenodo.7749239, https://doi.org/10.5281/zenodo.7749362, https://doi.org/10.5281/zenodo.7748792, https://zenodo.org/record/7860751 (accessed on 3 May 2023).

**Acknowledgments:** This study was developed in the context of the Program of Grant Funding for scientific and (or) scientific and technical projects for 2021–2023, supported by the Science Committee of the Ministry of Science and Higher Education of the Republic of Kazakhstan (grant number AP09259379).

**Conflicts of Interest:** The authors declare no conflict of interest.

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
