# Peer review of "A Multispectral UAV Imagery Dataset of Wheat, Soybean and Barley Crops in East Kazakhstan"

_data, 2023_

Round 1

Reviewer 1 Report

11/04/2023

Dear authors,

In the manuscript A multispectral UAV Imagery dataset of wheat, soybean, and barley crops in East Kazakhstan you introduce a dataset of crop imagery captured during the 2022 growing season in the Eastern Kazakhstan region, which were acquired using a multispectral camera mounted on an unmanned aerial vehicle, DJI Phantom 4. This facilitated thorough monitoring of the most important phenological stages of crop development in the experimental design, which consisted of 27 plots, each covering one hectare. The collected imagery underwent enhancement and expansion, integrating a sixth band that embodies the normalized difference vegetation index (NDVI) values, in conjunction with the original five multispectral bands. This amplification enables a more effective evaluation of vegetation health and growth, rendering the enriched dataset a valuable resource for the progression and validation of crop monitoring and yield prediction models, as well as for the exploration of precision agriculture methodologies.

General comments

Such data sets can be interesting and useful only to the authors and owner (farmer). Namely, authors collected imagery of certain fields and that data can only be used by the owner of the property. In authors case, they should emphasize the way of recording and preparing the data set for processing. It would have some relevance to a wider audience, as it is only a recorded set of data for a specific micro-area, and is not usable for the wider community.

In the Summary, authors should state how similar data collection is carried out and highlight the benefit of their method of collecting images with a drone.

If you would like to publish this manuscript in a some serious scientific journal (like this), it is necessary to specify specifics in the collection of such data that are not known until now, that is, they have not been used in such procedures. Otherwise, this is just a technical report from the data collecting.

Specific comments (are in the manuscript)

-          Line 37 - When you do serious scientific research, you must not stick to just one publishing group. In this way, you miss out on all other research, which is not professional and not right. I suggest you drop the naming of MDPI in the text. You don't need to pander to them to get your manuscript published.

-          Line 45 - Such manuscripts should be written in the third person. Change this throughout the text, please.

-          Line 49-52 – Do you really believe this is true? In today's age of numerous drones and sensors, when does a term like 'precision agriculture' exist!?

-          Line 61-66 - These are just speculations and have no place here. You should provide similar text with clear explanations in the User Notes.

-          Line 99-100 - What do you want to show or emphasize by displaying these images?

-          Line 209 - What do you mean by this? Explain in more details.

Best regards

Author Response

Response to Reviewer 1 Comments

11/04/2023

Dear authors,

In the manuscript A multispectral UAV Imagery dataset of wheat, soybean, and barley crops in East Kazakhstan you introduce a dataset of crop imagery captured during the 2022 growing season in the Eastern Kazakhstan region, which were acquired using a multispectral camera mounted on an unmanned aerial vehicle, DJI Phantom 4. This facilitated thorough monitoring of the most important phenological stages of crop development in the experimental design, which consisted of 27 plots, each covering one hectare. The collected imagery underwent enhancement and expansion, integrating a sixth band that embodies the normalized difference vegetation index (NDVI) values, in conjunction with the original five multispectral bands. This amplification enables a more effective evaluation of vegetation health and growth, rendering the enriched dataset a valuable resource for the progression and validation of crop monitoring and yield prediction models, as well as for the exploration of precision agriculture methodologies.

General comments

Such data sets can be interesting and useful only to the authors and owner (farmer). Namely, authors collected imagery of certain fields and that data can only be used by the owner of the property. In authors case, they should emphasize the way of recording and preparing the data set for processing. It would have some relevance to a wider audience, as it is only a recorded set of data for a specific micro-area, and is not usable for the wider community.

In the Summary, authors should state how similar data collection is carried out and highlight the benefit of their method of collecting images with a drone.

If you would like to publish this manuscript in a some serious scientific journal (like this), it is necessary to specify specifics in the collection of such data that are not known until now, that is, they have not been used in such procedures. Otherwise, this is just a technical report from the data collecting.

Response to General comments:

Dear Reviewer,

Thank you for your valuable feedback on our manuscript. We appreciate your comments and have carefully considered them in revising our work.

  • Regarding the relevance and significance of our data, we believe that our dataset has the potential to be useful to other researchers and stakeholders beyond our own research. The dataset covers three key crops for food security (wheat, barley, and soybean) over a growing season in the Eastern Kazakhstan region, providing an understanding of typical crop production in this vast area. The land area of the East Kazakhstan region is 283,226 square kilometers, which is larger than the land area of some European agricultural countries such as Bulgaria, the Netherlands, Greece, Romania, or the Czech Republic. However, the region belongs to the zone of risky farming due to various factors such as climate conditions, soil fertility, and other environmental factors. Agricultural activities in this region require special management and practices to minimize risks and ensure sustainable production. Therefore, our dataset can be used to identify trends and patterns in crop production in the zone of risky agriculture, and to compare agricultural cases with other regions. Although the dataset has a relatively small size of 27 hectares, it is very typical for the soil, climate, and agrotechnical conditions in East Kazakhstan. Therefore, we can say that our dataset is representative of the region. Furthermore, the dataset could be of interest to researchers from countries with similar climate and soil conditions, especially those belonging to the Transborder Altay region such as Mongolia, Siberia (Russian Federation), and western China.
  • Regarding highlighting the unique aspects of our data collection process, we agree that it is essential to emphasize the way we recorded and prepared the dataset. Although we provided a description of our data capturing and processing methods in Section 3, we understand that this information may not be sufficient. Therefore, we have highlighted the benefits of the drone-based approach for collecting crop imagery compared to satellite-based methods in the Summary (lines 33-42), as you suggested. Also, in line with your comment, we have added information in the Summary (lines 43-64) regarding how similar data was collected and processed and have highlighted that data collection methods are generally comparable. Therefore, we would like to emphasize that our contribution does not lie in the uniqueness of the data collection process, which is largely determined by the technical characteristics of the drone. Rather, our contribution lies in providing a set of raw data that other researchers can use to test their hypotheses and develop new methods, especially those who do not have access to their own crop imagery. We believe that our goal aligns with the aim of "Data", which emphasizes the significance of research data for scientific progress yet notes that most research data are not publicly available. As the journal has two sections - the Methods section for scientific and scholarly data collection, processing, management, storage, and analysis, and the Data Descriptors section for scientific and scholarly dataset descriptions - we intend to publish our findings in the latter section, focusing on the data description rather than specifics in the collection methods.

Specific comments

Point 1.

Line 37 - When you do serious scientific research, you must not stick to just one publishing group. In this way, you miss out on all other research, which is not professional and not right. I suggest you drop the naming of MDPI in the text. You don't need to pander to them to get your manuscript published.

Response to Point 1.

Thank you for your valuable comment. We would like to address your concern regarding the mention of only one publishing group. Please be assured that our literature review was not limited to just MDPI, as we also conducted a search in Elsevier's "Data in Brief" journal using the keywords "UAV" and "crop" (lines 45-47 in the first version of the manuscript). However, we fully agree with your point about the importance of considering multiple data sources when conducting a literature review. As a result, we have added two additional academic data sources to our search to ensure a comprehensive examination of the available literature.

Thus, in the end, we searched 4 data journals, namely: 1) Data (MDPI); 2) Data in Brief (Elsevier); 3) Earth System Science Data (Copernicus); 4) Scientific Data (Nature Research). These journals belong to different publishing groups, ensuring a comprehensive examination of the available literature. To avoid dependency on the search engine of each publishing house and standardize the search, we conducted our search on a single platform, Web of Science and described results (lines 65-85). We believe that these additions have strengthened our findings and provide a more robust basis for our conclusions. We hope that this clarifies any misunderstandings and thank you again for your valuable input.

Point 2.

Line 45 - Such manuscripts should be written in the third person. Change this throughout the text, please.

Response to Point 2.

Thank you for your valuable comment. We have made the necessary changes throughout the manuscript to ensure that it is written in the third person.

Point 3.

Line 49-52 – Do you really believe this is true? In today's age of numerous drones and sensors, when does a term like 'precision agriculture' exist!?

Response to Point 3.

Thank you very much. We have removed these lines from the manuscript. However, we still believe that not all researchers and academic communities have embraced the era of multispectral drones and sensors. Unfortunately, the concept of digital inequality is still present even in the academic environment.  

Point 4.

Line 61-66 - These are just speculations and have no place here. You should provide similar text with clear explanations in the User Notes.

Response to Point 4.

Thank you for your comment. We have toned down the language in our statements and removed some of them. However, we have left a small sentence, "It is expected that the dataset will be used to develop and test new algorithms and models for crop monitoring, disease detection, yield estimation, and other applications", as the journal guidelines for authors state that in the Summary section, "authors may wish to describe potential benefits of publicly releasing and describing the dataset”.

Point 5.

Line 99-100 - What do you want to show or emphasize by displaying these images?

Response to Point 5.

Thank you for your comment. We had intended to provide users of our dataset with sample images, but upon reflection, we agree that the images from any of the bands are not particularly illustrative. Therefore, we have decided to retain the first RGB image of the plot and add the image of the same plot, which was captured during a different flight session (a different phenological stage).

Point 6.

Line 209 - What do you mean by this? Explain in more details.

Response to Point 6.

Thank you for your comment. We added the explanation of Figure 8 to the text (lines 222-228).

Reviewer 2 Report

The manuscript generally fulfils the requirements required from the authors by the journal Data. However, the manuscript has some flaws in the organization and quality of writing. Certainly, the manuscript would benefit greatly from a review of the quality of English usage.

Section 1. Summary must be renamed Introduction and rewritten according to a suitable structure. The Introduction should better describe the context of the dataset and briefly and clearly give some insight into the data that are the object of the manuscript.

The description of the data should be more organic, clear and with more detail as explained in the instructions for authors in the journal of Data. In practice, even a non-expert reader should be able to understand how the data were collected and compiled.

Section 3 Methods, should be moved before the data description for a better understanding of the dataset. Also the methods should be better explained with more details.

Author Response

Response to Reviewer 2 Comments

22 Apr 2023 22:08:09

The manuscript generally fulfils the requirements required from the authors by the journal Data. However, the manuscript has some flaws in the organization and quality of writing. Certainly, the manuscript would benefit greatly from a review of the quality of English usage.

Section 1. Summary must be renamed Introduction and rewritten according to a suitable structure. The Introduction should better describe the context of the dataset and briefly and clearly give some insight into the data that are the object of the manuscript.

 The description of the data should be more organic, clear and with more detail as explained in the instructions for authors in the journal of Data. In practice, even a non-expert reader should be able to understand how the data were collected and compiled.

Section 3 Methods, should be moved before the data description for a better understanding of the dataset. Also the methods should be better explained with more details.

Response to Reviewer

Dear Reviewer,

Thank you for your comments on our manuscript. We appreciate your feedback and have taken it into consideration.

Regarding the renaming of the "Summary" section to "Introduction" and the suggestion to move the "Methods" section before the data description.

Thank you for your feedback on our article. While we appreciate your suggestion to improve the readability of our article, we kindly ask you to allow us to maintain our current structure.

We followed the guidelines of the "Data" journal for the data descriptor structure (https://www.mdpi.com/journal/data/instructions#manuscript) and strictly adhered to the recommended format. According to the guidelines, a Data Descriptor manuscript’s structure should be as follows:

  • Front matter: Title, Author list, Affiliations, Abstract, Keywords.
  • Dataset: title, DOI (or unique identifier), creator, publisher and location (e.g. URL) of the dataset
  • Dataset License
  • Research sections: Summary, Data Description, Methods, User notes
  • Back matter: Acknowledgments, Author Contributions, Conflicts of Interest, References.

Before submitting our dataset to the "Data" journal, we reviewed numerous publications and found that all of them followed the journal's guidelines for the breakdown of articles submitted as Data Descriptors. We appreciate your time and consideration and remain open to any further suggestions or comments you may have.

Regarding the better description of the context of the dataset and giving some insight into the data.

Dear Reviewer,

Thank you for your comment regarding the need for a more detailed and clear description of the data, as outlined in the instructions for authors in the Journal of Data. We have revisited the guidelines and addressed the following questions:

  1. What data is contained? This is answered in lines 96-107 of our manuscript.
  2. Which format? We have addressed this in lines 116-119 and 132-139.

To address the third question, "How can data be read and interpreted?", we have thoroughly revised Section 4 of our manuscript. Your comment has provided us with valuable feedback, enabling us to rethink the results of our work in a new way. We are grateful for this opportunity to improve our research.

Thank you again for your input, and please let us know if you have any further suggestions or feedback.

Reviewer 3 Report

The manuscript describes a dataset containing multispectral images captured using UAV. The text is well written and the dataset should be useful to the community. I have only two suggestions:

1. Does the dataset storage and availability adhere to the FAIR principles, which aim at improve the Findability, Accessibility, Interoperability, and Reuse of digital assets? If so, please state this explicitly in the manuscript. If not, I would recommend that the authors make an effort to meet those principles.

2. Some more detail about the conditions under which the images were captured would be useful (for example, weather conditions, time of the year, angle of insolation, etc.).

Author Response

Response to Reviewer 3 Comments

17 April 2023

The manuscript describes a dataset containing multispectral images captured using UAV. The text is well written, and the dataset should be useful to the community. I have only two suggestions:

  1. Does the dataset storage and availability adhere to the FAIR principles, which aim at improve the Findability, Accessibility, Interoperability, and Reuse of digital assets? If so, please state this explicitly in the manuscript. If not, I would recommend that the authors make an effort to meet those principles.
  2. Some more detail about the conditions under which the images were captured would be useful (for example, weather conditions, time of the year, angle of insolation, etc.).

Response to comments:

Dear Reviewer,

Thank you for your valuable feedback on our manuscript. We appreciate your comments and have carefully considered them in revising our work.

Response to Point 1.

We have updated our manuscript to explicitly state that our dataset adheres to the FAIR principles (lines 265-275). We appreciate your valuable feedback and believe that this addition strengthens the quality and impact of our work.

Response to Point 2.

We have added this information to the manuscript as Table 4. We appreciate your valuable feedback.

Round 2

Reviewer 1 Report

29/04/2023

Dear authors,

In the manuscript A multispectral UAV Imagery dataset of wheat, soybean, and barley crops in East Kazakhstan you introduce a dataset of crop imagery captured during the 2022 growing season in the Eastern Kazakhstan region, which were acquired using a multispectral camera mounted on an unmanned aerial vehicle, DJI Phantom 4. This facilitated thorough monitoring of the most important phenological stages of crop development in the experimental design, which consisted of 27 plots, each covering one hectare. The collected imagery underwent enhancement and expansion, integrating a sixth band that embodies the normalized difference vegetation index (NDVI) values, in conjunction with the original five multispectral bands. This amplification enables a more effective evaluation of vegetation health and growth, rendering the enriched dataset a valuable resource for the progression and validation of crop monitoring and yield prediction models, as well as for the exploration of precision agriculture methodologies.

Response to General comments:

Regarding the relevance and significance of our data, we believe that our dataset has the potential to be useful to other researchers and stakeholders beyond our own research. The dataset covers three key crops for food security (wheat, barley, and soybean) over a growing season in the Eastern Kazakhstan region, providing an understanding of typical crop production in this vast area. The land area of the East Kazakhstan region is 283,226 square kilometers, which is larger than the land area of some European agricultural countries such as Bulgaria, the Netherlands, Greece, Romania, or the Czech Republic. However, the region belongs to the zone of risky farming due to various factors such as climate conditions, soil fertility, and other environmental factors. Agricultural activities in this region require special management and practices to minimize risks and ensure sustainable production. Therefore, our dataset can be used to identify trends and patterns in crop production in the zone of risky agriculture, and to compare agricultural cases with other regions. Although the dataset has a relatively small size of 27 hectares, it is very typical for the soil, climate, and agrotechnical conditions in East Kazakhstan. Therefore, we can say that our dataset is representative of the region. Furthermore, the dataset could be of interest to researchers from countries with similar climate and soil conditions, especially those belonging to the Transborder Altay region such as Mongolia, Siberia (Russian Federation), and western China.

Regarding highlighting the unique aspects of our data collection process, we agree that it is essential to emphasize the way we recorded and prepared the dataset. Although we provided a description of our data capturing and processing methods in Section 3, we understand that this information may not be sufficient. Therefore, we have highlighted the benefits of the drone-based approach for collecting crop imagery compared to satellite-based methods in the Summary (lines 33-42), as you suggested. Also, in line with your comment, we have added information in the Summary (lines 43-64) regarding how similar data was collected and processed and have highlighted that data collection methods are generally comparable. Therefore, we would like to emphasize that our contribution does not lie in the uniqueness of the data collection process, which is largely determined by the technical characteristics of the drone. Rather, our contribution lies in providing a set of raw data that other researchers can use to test their hypotheses and develop new methods, especially those who do not have access to their own crop imagery. We believe that our goal aligns with the aim of "Data", which emphasizes the significance of research data for scientific progress yet notes that most research data are not publicly available. As the journal has two sections - the Methods section for scientific and scholarly data collection, processing, management, storage, and analysis, and the Data Descriptors section for scientific and scholarly dataset descriptions - we intend to publish our findings in the latter section, focusing on the data description rather than specifics in the collection methods.

Specific comments

Point 1.

Line 37 - When you do serious scientific research, you must not stick to just one publishing group. In this way, you miss out on all other research, which is not professional and not right. I suggest you drop the naming of MDPI in the text. You don't need to pander to them to get your manuscript published.

Response to Point 1.

Thank you for your valuable comment. We would like to address your concern regarding the mention of only one publishing group. Please be assured that our literature review was not limited to just MDPI, as we also conducted a search in Elsevier's "Data in Brief" journal using the keywords "UAV" and "crop" (lines 45-47 in the first version of the manuscript). However, we fully agree with your point about the importance of considering multiple data sources when conducting a literature review. As a result, we have added two additional academic data sources to our search to ensure a comprehensive examination of the available literature.

Thus, in the end, we searched 4 data journals, namely: 1) Data (MDPI); 2) Data in Brief (Elsevier); 3) Earth System Science Data (Copernicus); 4) Scientific Data (Nature Research). These journals belong to different publishing groups, ensuring a comprehensive examination of the available literature. To avoid dependency on the search engine of each publishing house and standardize the search, we conducted our search on a single platform, Web of Science and described results (lines 65-85). We believe that these additions have strengthened our findings and provide a more robust basis for our conclusions. We hope that this clarifies any misunderstandings and thank you again for your valuable input.

Point 2.

Line 45 - Such manuscripts should be written in the third person. Change this throughout the text, please.

Response to Point 2.

Thank you for your valuable comment. We have made the necessary changes throughout the manuscript to ensure that it is written in the third person.

Point 3.

Line 49-52 – Do you really believe this is true? In today's age of numerous drones and sensors, when does a term like 'precision agriculture' exist!?

Response to Point 3.

Thank you very much. We have removed these lines from the manuscript. However, we still believe that not all researchers and academic communities have embraced the era of multispectral drones and sensors. Unfortunately, the concept of digital inequality is still present even in the academic environment.

Point 4.

Line 61-66 - These are just speculations and have no place here. You should provide similar text with clear explanations in the User Notes.

Response to Point 4.

Thank you for your comment. We have toned down the language in our statements and removed some of them. However, we have left a small sentence, "It is expected that the dataset will be used to develop and test new algorithms and models for crop monitoring, disease detection, yield estimation, and other applications", as the journal guidelines for authors state that in the Summary section, "authors may wish to describe potential benefits of publicly releasing and describing the dataset”.

Point 5.

Line 99-100 - What do you want to show or emphasize by displaying these images?

Response to Point 5.

Thank you for your comment. We had intended to provide users of our dataset with sample images, but upon reflection, we agree that the images from any of the bands are not particularly illustrative. Therefore, we have decided to retain the first RGB image of the plot and add the image of the same plot, which was captured during a different flight session (a different phenological stage).

Point 6.

Line 209 - What do you mean by this? Explain in more details.

Response to Point 6.

Thank you for your comment. We added the explanation of Figure 8 to the text (lines 222-228).

General comments

You answered all my comments and questions. I am mostly satisfied with them and have no major complaints.

Best regards

Reviewer 2 Report

The manuscript has been sufficiently revised and my comments adequately addressed. As far as I am concerned, the manuscript is now acceptable for publication.